# Isotherm, Kinetic, and Selectivity Studies for the Removal of ^133^Ba and ^137^Cs from Aqueous Solution Using Turkish Perlite

**DOI:** 10.3390/ma15217816

**Published:** 2022-11-05

**Authors:** Süleyman İnan, Vipul Vilas Kusumkar, Michal Galamboš, Eva Viglašová, Oľga Rosskopfová, Martin Daňo

**Affiliations:** 1Institute of Nuclear Sciences, Ege University, Bornova, 35100 İzmir, Türkiye; 2Department of Nuclear Chemistry, Faculty of Natural Sciences, Comenius University in Bratislava, Ilkovicova 6, 842 15 Bratislava, Slovakia; 3Department of Nuclear Chemistry, Faculty of Nuclear Sciences and Physical Engineering, Czech Technical University in Prague, Břehová 7, 115 19 Prague, Czech Republic

**Keywords:** cesium, barium, sorption, perlite, radioactive waste

## Abstract

The efficiency of ${}^{133}\mathrm{Ba}$ and ${\;}^{137}\mathrm{Cs}$ removal from aqueous solution is vital to mitigate ecological concerns over spreading these radionuclides in the environment. The present work focused on the use of Turkish perlite for the sorptive removal of ${\;}^{133}\mathrm{Ba}$ and ${\;}^{137}\mathrm{Cs}$ from aqueous solution by the radioindicator method. Perlite was characterized by XRF, XRD, FTIR, SEM–EDX, and BET analyses. The maximum percentage removals of 88.2% and 78.7% were obtained for ${\;}^{133}\mathrm{Ba}$ and ${\;}^{137}\mathrm{Cs}$ at pH 6 and pH 9, respectively. For both ions, the sorption equilibrium was attained relatively rapidly. Experimental kinetic data were well described with pseudo-second-order and intraparticle diffusion models. The uptake of both ions increased with the increase in metal concentration (1 × 10^−5^ to 5 × 10^−2^ mol/L) in solution. The maximum uptake capacities of ${\;}^{133}\mathrm{Ba}$ and ${\;}^{137}\mathrm{Cs}$ were found to be 1.96 and 2.11 mmol/g, respectively. The effect of competing ions decreased in the order of ${\mathrm{Ca}}^{2+}>K^{+}>{\mathrm{Ni}}^{2+}>{\mathrm{Na}}^{+}$ for ${\;}^{133}\mathrm{Ba}$ sorption, whereas for ${\;}^{137}\mathrm{Cs}$ sorption, the order was determined as ${\mathrm{Ca}}^{2+}>{\mathrm{Ni}}^{2+}>K^{+}>{\mathrm{Na}}^{+}$. Selectivity studies pointed out that sorption of ${\;}^{133}\mathrm{Ba}$ onto perlite is preferable to ${\;}^{137}\mathrm{Cs}$. Therefore, Turkish perlite is a promising, cost-effective, and efficient natural material for the removal of ${\;}^{133}\mathrm{Ba}$ and ${\;}^{137}\mathrm{Cs}$ from relatively diluted aqueous solution.

## 1. Introduction

${\;}^{133}\mathrm{Ba}$ and C137s are both released into the environment via testing of nuclear weapons, via effluents release containing radioactive waste from nuclear power plants (NPPs), processing operations of nuclear fuel, and accidents in NPPs [1]. Barium and cesium have many isotopes, but only some of them are important and possess major concern for the environment due to their longer half-life. In the case of C137s ~ 30.08 years (β^−^ emitter) and for B133a ~ 10.55 years (γ emitter), both are generated during fission of uranium and plutonium. Barium and cesium salts are highly soluble in water, which improves their distribution in the soil. Both ions are not essential elements for plants and animals; however, they are known for their toxic effect on them. Plants can accumulate B133a and C137s from soil if adequately solubilized, helping them to enter the food chain and resulting in harmful effects on animals and on plant species due to the internal exposure of β and γ radiation caused by decaying radionuclides [2,3]. Concentration limits for C137s in drinking water, milk, and food are provided in Table 1.

Therefore, the removal of potentially hazardous radionuclides from liquid waste streams is a very important issue to prevent possible effects on living species and the environment. There have been many different methods applied to remove barium and cesium from the wastewater, mainly phytoremediation [7], ion-exchange [8], the ion floatation process [9], chemical precipitation [10], solvent extraction [11,12], electrochemical purification [13], membrane separation [14], and adsorption [15]. Adsorption is considered the most useful method for the barium and cesium removal and has been widely used. Various types of synthetic and natural materials have been proposed for the adsorption of barium and cesium from aqueous solution and waste streams. Natural inorganic materials have been preferred because of their abundance, low cost, radiation resistance, and reasonable capacity for radionuclides. Among these, the most reported materials are zeolites [16,17,18], clay minerals [19,20,21], rocks [22,23], and fly ash [24]. On the other hand, perlite is another interesting naturally occurring mineral composed of amorphous alumina-silicate volcanic rock belonging to the granite-rhyolite family under the sub-group of glassy rocks [25]. Perlite mine locations are in different regions of the world. Based on the estimated world production for 2021, the world’s leading producers were China, Turkey, Greece, and the United States. It was estimated that there are approximately 4200 thousand metric tons of perlite produced around the world during 2021 [26].

Due to a certain amount of water in its internal structure, it has the capacity to expand up to 4–20 times when heated at high temperature (760–1100 °C). By expansion, perlite becomes very light and porous. Perlite rocks can have different properties in terms of color and structure. The color of raw perlite can vary from transparent light gray to bright black. The most important feature of perlite is the water it contains as a compound of 2.5% in the hydrated glassy silica structure, and this water ensures the stability of the perlite [27]. It is abundantly available and cheap in Turkish markets. Its high porosity and permeability, large surface area, and ultra-light weight make perlite a suitable sorbent material for the uptake of organic pollutants [28,29], heavy metal ions [30,31,32,33], and radionuclides [34,35] from aqueous waste streams.

Talip et al. (2009) conducted batch tests to examine the effect of parameters for thorium sorption on expanded perlite as a low-cost material. They reported that the most dominant factors were solution pH and initial thorium concentration. Thorium sorption was investigated between pH 2 and 7 and the maximum thorium sorption (%) was obtained at pH 4.5. By the investigation of thorium concentration in the range of 20–250 mg/L, a sorption uptake (%) of 84 ± 4 was found at 50 mg/L concentration [34]. In another study, the removal of Sr and Ba onto Iranian expanded perlite was reported by Torab-Mostaedi et al. (2011). pH, sorbent dosage, contact time, and temperature were the affecting parameters under investigation. They said that the pH was the most significant factor on Sr and Ba adsorption, and adsorption (%) was maximized at pH 6 for both ions. From the isotherms, it can be deduced that the adsorption is Langmuir type and the monolayer coverages of Sr and Ba ions were determined to be 1.14 and 2.486 mg/g, respectively [35]. Cabranes et al. (2018) examined the use of Argentinian perlite for the removal of radioactive cesium from wastewater. Although the surface area of the perlites was low, it was observed that their capacity for Cs^+^ was good. They reported that the increase in pH and modification with NaOH resulted in a higher percentage removal. The Cs^+^ percentage removal of perlite obtained from Pava mine increased from 84 to 89% after NaOH treatment when the sorbent dosage and initial metal concentration were maintained at 30 g/L and 10 mg/L, respectively. The maximum Cs^+^ adsorption capacity of Pava was found to be 2.91 mg/g [36].

There are very few studies reported in the literature focused on the sorption of fission products by perlite-based materials, and inactive ions were used in almost all these studies. In addition, utilization of Turkish perlite as a sorbent for B133a and C137s has not been reported yet.

In the present study, we focused on the characterization studies of Turkish perlite and its sorption behaviors toward B133a and C137s from diluted aqueous solution. The influences of pH, contact time, metal concentration, and dosage on sorption were investigated. Sorption properties of perlite in the presence of competing ions were examined and a selectivity study was conducted.

## 2. Materials and Methods

### 2.1. Materials

Perlite samples were supplied from Harborlite Aegean End. Min. San. A.Ş. (Bergama-İzmir, Turkey). Following the extraction from the mine, perlite was subjected to pre-crushing, drying, crushing, sieving, and classification processes. The particle size distribution was in the range of 0.075–0.6 mm. All chemicals used in experiments were of analytical grade and supplied from Slavus, s.r.o, Bratislava, Slovak Republic, or Lacherna n.p., Brno, Czech Republic. Radiotracer concentrations were as follows: ${\;}^{133}\mathrm{Ba}$ in the form of [${\;}^{133}\mathrm{Ba}$]BaCl_2_ (Eurostandard CZ s.r.o., Czech Republic) with a volume activity of 2 MBq/mL and ${\;}^{137}\mathrm{Cs}$ in the form of [${\;}^{137}\mathrm{Cs}$]CsCl (National Centre for Nuclear Research, Poland) with a volume activity of 55 MBq/mL.

### 2.2. Sorption Experiments

The sorption of Ba and Cs on Turkish perlite was performed by the radioisotope indication method using a radioisotope of ${\;}^{133}\mathrm{Ba}$ and ${\;}^{137}\mathrm{Cs}$. Experiments were carried out by the batch method under aerobic conditions and at the laboratory temperature. The parameters affecting sorption were examined by adding 5 mL of aqueous phase to 0.05 g of perlite in a plastic tube. Perlite and aqueous phase were mixed in a rotary laboratory mixer with a constant speed of mixing. After sorption, the suspension was centrifuged at 8000 rot min^−1^ for 15 min and an aliquot of each supernatant was collected and analyzed with a Modumatic model gamma spectrometer equipped with a NaI(Tl) detector. The statistical error of the measurement was below ~1%.

The effect of initial pH on Ba and Cs sorption was investigated between pH 3 and 9. The influence of contact time was studied from 1 min up to 180 min. Isotherm studies were carried out using a solution of Ba^2+^ or Cs^+^, with initial concentrations of 1 × 10^−^^5^ to 5 × 10^−^^2^ mol/L. The effect of sorbent dosage on sorption was examined from 5 to 80 g/L.

For competitive ion studies, Na^+^, K^+^, Ni^2+^, and Ca^2+^ were chosen as competing ions and binary solutions of these ions with Ba^2+^ and Cs^+^. Each of these solutions was prepared at pH 6 and 9, respectively. In the solutions, the concentration of Ba^2+^ and Cs^+^ ions was fixed to 1 × 10^−^^5^ mol/L, whereas concentrations of Na^+^, K^+^, Ni^2+^, and Ca^2+^ as competing ions were increased from 1 × 10^−^^5^ to 5 × 10^−^^2^ mol/L. An amount of 0.05 g of perlite was contacted with each binary metal containing solution for 60 min.

Selectivity studies were performed by contacting 0.05 g of perlite with equimolar (1 × 10^−^^5^ mol/L) Ba^2^- and Cs^+^-bearing solutions within the pH range of 3–9, for 60 min.

Sorption properties of perlite toward Ba and Cs were calculated using the following equations:Distribution coefficient (*K_d_*)
(1)$$K_{d}=\frac{\left(a_{0}-a\right)}{a}\times \frac{V}{m}\;\left(\mathrm{mL}/g\right)$$Sorption percentage (*R*)
(2)$$R=\left(100\times K_{d}\right)/\left(K_{d}+V/m\right)\;\left(\%\right)$$Sorption capacity (*q*)
(3)$$q=K_{d}\times C_{eq}\;\left(\mathrm{mmol}/g\right)$$Equilibrium concentration (*C_eq_*)
(4)$$C_{eq}=\frac{C_{0}\times a}{a_{0}}\;(\mathrm{mol}/L)$$Selectivity coefficient (*β*)
(5)$$\beta_{1,2}=\frac{K_{d1}}{K_{d2}}$$
where *C*_0_ and *C_eq_* are the initial and equilibrium concentration (mol/L), respectively, *V* is the volume of aqueous phase (mL), *m* is the mass of sorbent (g), *a* and *a*_0_ are the initial and equilibrium volume activities of the solutions (mL/s), respectively, and *K_d_*_1_ and *K_d_*_2_ are the distribution coefficients of ${\;}^{133}\mathrm{Ba}$ and ${\;}^{137}\mathrm{Cs}$, respectively.

### 2.3. Characterization Studies

X-ray fluorescence analysis (XRF) was carried out for the purpose of perlite’s chemical composition identification. X-ray diffraction (XRD) analysis was used to evaluate the crystallinity and structure of perlite. The sample was measured on a MiniFlex 600 XRD diffractometer (Rigaku, Tokyo, Japan) in range of 5°–80° at the velocity of 2° min^−1^. Fourier Transform Infrared Spectroscopy (FTIR) spectra were acquired between 400 and 4000 cm^−^^1^ by a Perkin Elmer Spectrum Two model FTIR-ATR instrument (Waltham, MA, USA). Brunauer–Emmett–Teller (BET) surface area analysis provided data on the textural properties of the sorbent material. BET surface area measurement was carried out using the MONOSORB™ MS-22 device (Quantachrome Instruments, Boynton Beach, FL, USA) by the one-point measurement method with a working gas of 30 mol. % N_2_ and 70 mol. % He. Scanning electron microscopy–energy-dispersive X-ray (SEM–EDX) analysis identified the morphological structure of the sorbent surface and gave an idea on the elemental composition of the perlite surface. Scanning electron microscopy and EDX analysis of the perlite sample were performed using a VEGA 2 SEM microscope (TESCAN s.r.o., Brno, Czech Republic) coupled with a QUANTAX QX2 EDX detector (RONTEC, Denkendorf, Germany). Before SEM–EDX analysis, the perlite sample was fixed on an aluminum sample holder using conductive adhesive (Ag). The sample was then coated with Au using a vacuum coating system (TESLA ELMI a.s., Brno, Czech Republic). The analysis was carried out at 500× magnification, a pressure of 36 × 10^−^^3^ Pa, and a voltage of 30 kV.

## 3. Results and Discussion

### 3.1. Characterization of Perlite

The elemental composition (identification only) of Turkish perlite was examined by XRF analysis. The most occurring elements were Si, Al, K, Fe, Ni, As, Zn, Rb, Mn, or Zr. This identification proved that the composition of perlite is similar to other perlites used in the literature [32,33].

XRD data of perlite are shown in Figure 1a. There are two amorphous peaks in the spectrum, one narrower—around 2θ ≈ 4°, and one wider with a peak around 2θ ≈ 25°. These two peaks probably correspond to the structure of amorphous volcanic glass. Identifying sharp peaks is more difficult. The closest possible structures that could belong to a given peak are: quartz and polylithionite. The FTIR spectrum of perlite is illustrated in Figure 1b. The bands at 3620 and 1730 cm^−^^1^ are indicators of O-H stretching and bending vibrations. The sharp peak at 1013 cm^−^^1^ represents the Si-O bond. The peaks around 775 and 707 cm^−^^1^ can be attributed to the stretching of Si-O-Si and bending of Si-O-Al groups, respectively [37]. The BET surface area of perlite was determined to be 1.4 m^2^/g. Similar values were reported by Torab-Mostaedi et al. [35] and Alkan and Doğan [38]. The SEM–EDX analysis of perlite is presented in Figure 1c. The SEM image illustrates the porous structure of the perlite surface. As expected in perlite, Si, Al, O, K, Mg, Na, Ca, and Fe elements were determined and the data corresponded with the data provided by XRF analysis.

### 3.2. Sorption Studies

#### 3.2.1. Effect of pH

The surface of perlite is mainly covered by hydroxyl groups attached to silanol groups and alumina, which determines its sorption behaviors. Silanol groups and hydrous oxide surface groups in alumina are responsible for the adsorption of Ba^2+^ and Cs^2+^ onto perlite [29]. The schematic illustration of surface functional groups on perlite can be seen in Figure 2.

pH is a significant parameter affecting the sorption of metal ions onto solid surfaces. With the alteration of solution pH, protonation and deprotonation reactions occur and the surface of the material is positively or negatively charged. As the solution pH decreases, the surface of the sorbent becomes more positively charged and the sorption of metal ions decreases. Conversely, with an increase in pH, the surface becomes negatively charged and the sorption capacity increases. As shown in Figure 3, the ionization of silanol groups can be enhanced by the increase in solution pH and hydrogen ions are transferred from surface to solution. Thus, metal ions can be sorbed on negatively charged surfaces [39,40].

In our study, Ba and Cs sorption as a function of initial pH was investigated in the pH range of 3–9. Figure 4 shows that perlite had a higher sorption percentage for Ba^2+^ than Cs^+^ ions in the whole pH range. At pH 3, the sorption percentage of both ions was lower. The maximum sorption percentage was obtained to be 88.2% and 78.7% for Ba^2+^ and Cs^+^ at pH 6 and pH 9, respectively. Therefore, for further studies, the initial pH was fixed at pH 6 and pH 9 for Ba^2+^ and Cs^+^, respectively.

#### 3.2.2. Effect of Contact Time

The uptake capacity of Ba^2+^ and Cs^+^ as a function of contact time is shown in Figure 5. Perlite had a higher capacity for Ba^2+^ in comparison to Cs^+^ throughout the contact time range. The uptake capacity of both ions increased up to 30 min, then became almost constant. Further studies were conducted for 60 min to ensure the equilibrium was established. Sorption equilibrium was also maintained at 60 min for thorium and 90 min for cesium, barium, strontium, copper, and lead [34,35,36,41] in similar studies using perlite.

Pseudo-first-order, pseudo-second-order, and intraparticle diffusion kinetic models were applied to explain the experimental data. The pseudo-first-order equation [42] in its linear form is given as:(6)$$ln\left(q_{e}-q_{t}\right)=lnq_{e}-k_{1}\times t$$
where *q_t_* and *q_e_* are the amounts of metal ions sorbed at equilibrium (mg/g) and *t* (min), respectively, and *k*_1_ is the rate constant of the equation (min^−^^1^). The sorption rate constant (*k*_1_) represents the slope of the graph course drawn between *ln*(*q_e_-q_t_*) and *t*. The relatively low R^2^ values (0.14 for Ba^2+^ and 0.45 for Cs^+^ sorption) suggest that the sorption of these ions onto perlite does not fit a pseudo-first-order kinetic model (Table 2).

The pseudo-second-order model has been proposed to predict the kinetic behavior of adsorption where chemical sorption is the rate control step. The linearized form of the model is expressed as follows [43,44]:(7)$$\frac{t}{q_{t}}=\frac{1}{k_{2}\times q_{e}^{2}}+\left(\frac{1}{q_{e}}\right)\times t$$
where *k*_2_ is the second-order rate constant (g/mmol min); *q_t_* and *q_e_* (mmol/g) are the amount of metal adsorbed at adsorption time *t* (min) and equilibrium, respectively. The linear plots of *t*/*q_t_* versus *t* for the adsorption of Ba^2+^ and Cs^+^ ions onto perlite are shown in Figure 6a. The values of *k*_2_, *R*^2^, and *q_e_* are given in Table 2. The *R*^2^ values obtained were relatively high (0.999) for Ba^2+^ and Cs^+^ sorption. These data suggest that the sorption of Ba^2+^ and Cs^+^ ions onto perlite fits well with the pseudo-second-order kinetics. Similar results were also reported for the sorption of metal ions onto perlite-based sorbents [35,36,41,45].

Weber and Morris (1963) developed and proposed an intraparticle diffusion model, which is based on pore diffusion [46]. The linearized form of the model is as follows:(8)$$q_{t}=K_{i}\times t^{0.5}+C$$

According to the internal diffusion model, the internal diffusion of the adsorbate is the slowest step, and, thus, it is considered as the rate-determining step during the adsorption process [47]. Diffusion and adsorption are both influenced by surface area, reactivity of the surface, and reaction of the surface and pore structure for internal diffusion. Intraparticle diffusion is characterized by efficient mixing, a bigger particle size of adsorbent, high adsorbate concentration, and relatively low affinity of the adsorbate for the adsorbent. The model is defined by a plot expressing the relationship between metal uptake (*q_t_*) and the square root of time (*t*^1/2^), as shown in Figure 6b. The calculated model parameters are provided in Table 3. In Equation (8), the *C* value denotes the boundary layer thickness, and a higher *C* value means a thicker boundary layer [48,49,50]. If the value of *C* is zero, there is no boundary layer and the curve intercepts zero. As a result, film diffusion is negligible when the layer thickness is equal to zero, and, thus, intraparticle diffusion is considered as the rate controlling step. However, this is a theoretical approach related to Equation (8). Many adsorption studies have reported that both intraparticle and film diffusion are responsible for the rate-limiting step. As shown in Figure 6b and Table 3, experimental data were divided into two different sections for Ba^2+^ and Cs^+^ sorption. Figure 6b was divided by two linear regressions, demonstrating that adsorption diffusion is controlled by both film and intraparticle diffusion.

Comparing the data based on Table 3, *K_id_* values for the film diffusion and intraparticle diffusion sections for Ba^2+^ and Cs^+^ ions demonstrate that the rate limiting step was the intraparticle diffusion. This is because the intraparticle diffusion constant *K_id_*_2_ values for Ba^2+^ and Cs^+^ ions were 1 × 10^−6^ and 2 × 10^−6^, respectively, lower than those for the film diffusion constants *K_id_*_1_ of 5 × 10^−5^ and 3 × 10^−5^ for Ba^2+^ and Cs^+^ ions, respectively. This shows that the rate of intraparticle diffusion was slower and the rate determining step. Furthermore, in the first regression line, the *K_id_*_1_ value of Ba^2+^ was higher than that of Cs^+^, while the *K_id_*_2_ value of Cs^+^ was found to be higher in the second line. This means the transport of Ba^2+^ ions from solution through liquid film on the adsorbent surface was faster than that of Cs^+^ ions in the first part. Thereafter, in the second part, Cs^+^ ions filled the micropores faster than Ba^2+^ ions did.

#### 3.2.3. Effect of Metal Ion Concentration

The change in the uptake capacity of Ba^2+^ and Cs^+^ ions as a function of initial metal ion concentration was studied in the range of 1 × 10^−5^ and 5 × 10^−2^ mol/L. For both ions, similar isotherm curves were obtained. By the increase in the amount of metal ions in solution, the uptake capacity of the sorbent increased. When the initial concentration was equal to 5 × 10^−2^ mol/L, uptake capacities for Ba^2+^ and Cs^+^ were calculated as 1.96 and 2.11 mmol/g, respectively. Sorption isotherm curves of these cations are presented in Figure 7a.

Adsorption isotherms express the relationship between the concentration of adsorbed species and the degree of adsorption at a constant temperature. The Langmuir model assumes that the adsorption is monolayer and takes place at certain homogeneous regions on the adsorbent surface. The linearized form of the Langmuir equation can be written as follows [51]:(9)$$\frac{C_{e}}{q_{e}}=\frac{C_{e}}{q_{m}}+\left(\frac{1}{q_{m}\times b}\right)$$
where *q_e_* is the amount of metal ions adsorbed at equilibrium (mol/g), *C_e_* is the concentration of metal ions in the solution at equilibrium (mol/L), *q_m_* is the monolayer adsorption capacity (mol/g), and *b* is the constant related to the free energy of adsorption (L/g). The fitting of the experimental data to the Langmuir model for the sorption of both ions onto perlite was found to be poor based on the *R*^2^ values, as given in Table 4. This reveals that sorption on the surface did not occur as a monolayer coverage.

The Freundlich model [52] can be applied for multilayer sorption on heterogeneous surfaces. It is written in its linear form as
(10)$$logq_{e}=logK_{f}+\frac{1}{n}\times logC_{e}$$
where *K_f_* (mol/g) is a constant related to the adsorption capacity and 1/*n* is a parameter dependent on the adsorption intensity. It is clear from Figure 7b and Table 4 that Ba^2+^ and Cs^+^ sorption data fit the Freundlich model well. *K_f_* and 1/*n* values were calculated to be 0.015 and 0.71 for Ba^2+^ and 0.027 and 0.81 for Cs^+^, respectively. The 1/*n* values obtained between 0 and 1 indicated that Ba^2+^ and Cs^+^ sorption onto perlite was favorable under these working conditions.

Several types of natural inorganic sorbents have been used for the treatment of radioactive waste solutions. Ba^2+^ and Cs^+^ sorption capacities of different natural minerals are illustrated in Table 5. Turkish perlite has a reasonable sorption capacity in comparison with the other materials reported in literature.

#### 3.2.4. Effect of Dosage

The effect of sorbent dosage on the uptake capacity and sorption percentage of Ba^2+^ and Cs^+^ ions is shown in Figure 8. When the sorbent dosage was increased from 5 to 80 g/L, the sorption percentage raised from 43.2% to 72.7% for Ba^2+^ and from 54.1% to 83.0% for Cs^+^ ions.

On the contrary, with the increase in sorbent dosage, a gradual decrease can be seen on Ba^2+^ and Cs^+^ uptake capacity. The increase in the sorption percentage can be explained by the increase in the active sites on the sorbent and, thus, the easier penetration of metal ions into the sorption sites.

### 3.3. Effect of Competing Ions and Selectivity Studies

The metal ions to be removed in wastewater are found together with various similar ions. These ions can compete for the active binding sites on the sorbent surface. The influence of competitive cations (Na^+^, K^+^, Ni^2+^, and Ca^2+^) on the Ba^2+^ and Cs^+^ sorption was studied. It can be clearly seen from Figure 9a,b that the sorption of Ba^2+^ and Cs^+^ on perlite was gradually decreased with increasing concentration of competing ions in solution with concentrations from 1 × 10^−5^ to 5 × 10^−2^ mol/L. Ca^2+^ was the most competitive ion with both Ba^2+^ and Cs^+^. For Ba^2+^ sorption, the effect of competing ions was of the order of ${\mathrm{Ca}}^{2+}>K^{+}>{\mathrm{Ni}}^{2+}>{\mathrm{Na}}^{+}$, whereas for Cs^+^ sorption, the order was as follows ${\mathrm{Ca}}^{2+}>{\mathrm{Ni}}^{2+}>K^{+}>{\mathrm{Na}}^{+}$. On the other hand, even at the highest competing ion concentration of 5 × 10^−2^ mol/L, the sorption percentage of Ba^2+^ and Cs^+^ was determined to be above 40%.

Figure 10a shows the change in *K_d_* values of Ba^2+^ and Cs^+^ with the alteration of solution pH from 3 to 9. For both metal ions, a gradual increase was observed in *K_d_* values up to pH 7, after which a gradual decrease was observed. Maximum *K_d_* values for Ba^2+^ and Cs^+^ at pH 7 were determined as 960.4 and 418.6 mL/g, respectively.

When the variation in the selectivity coefficients depending on pH was examined (Figure 10b), it was found that perlite showed more selectivity for Ba^2+^ than Cs^+^ ions at each pH under investigation. Above pH 4, the selectivity coefficient (*β*) of Ba/Cs was higher than 2, which is an indicator of a good selectivity. The highest value of *β_Ba/Cs_* (2.69) was obtained at pH 6.

## 4. Conclusions

In this study, we present research on the use of Turkish perlite for the removal of B133a and C137s radioisotopes. Turkish perlite, a low-cost and an abundant mineral, exhibited promising potential for the remediation of contaminated solutions. Our study is the first focused on the removal of ^133^Ba and ^13^⁷Cs by Turkish perlite.

To understand the physical characteristics of perlite, XRF, XRD, FTIR, SEM–EDX, and BET analyses were carried out. Sorption tests performed in the range of pH 3–9 revealed that the sorption of Ba^2+^ and Cs^+^ was not remarkably affected by pH, both showing more than 75 and 60% of the uptake, respectively. The Freundlich isotherm model was the best fit for both Ba^2+^ and Cs^+^, which indicates heterogenous multilayer sorption with R^2^ values of 0.976 and 0.996, respectively. The sorption of Ba^2+^ and Cs^+^ on perlite followed pseudo-second-order and intraparticle diffusion kinetic models. The binary sorption studies were conducted in the presence of different cations, which showed a reduction in sorption percentage of Ba^2+^ and Cs^+^ in the order of ${\mathrm{Ca}}^{2+}>K^{+}>{\mathrm{Ni}}^{2+}>{\mathrm{Na}}^{+}$ and ${\mathrm{Ca}}^{2+}>{\mathrm{Ni}}^{2+}>K^{+}>{\mathrm{Na}}^{+}$, respectively. The selectivity study confirmed that Turkish perlite was selective toward Ba^2+^ ions and it had a *β* value of 2.69 for Ba/Cs at pH 6. Hence, Turkish perlite is proved to be a useful, efficient material for the removal of Ba^2+^ and Cs^+^ from relatively diluted aqueous solution.

## Figures and Tables

**Figure 1 materials-15-07816-f001:**
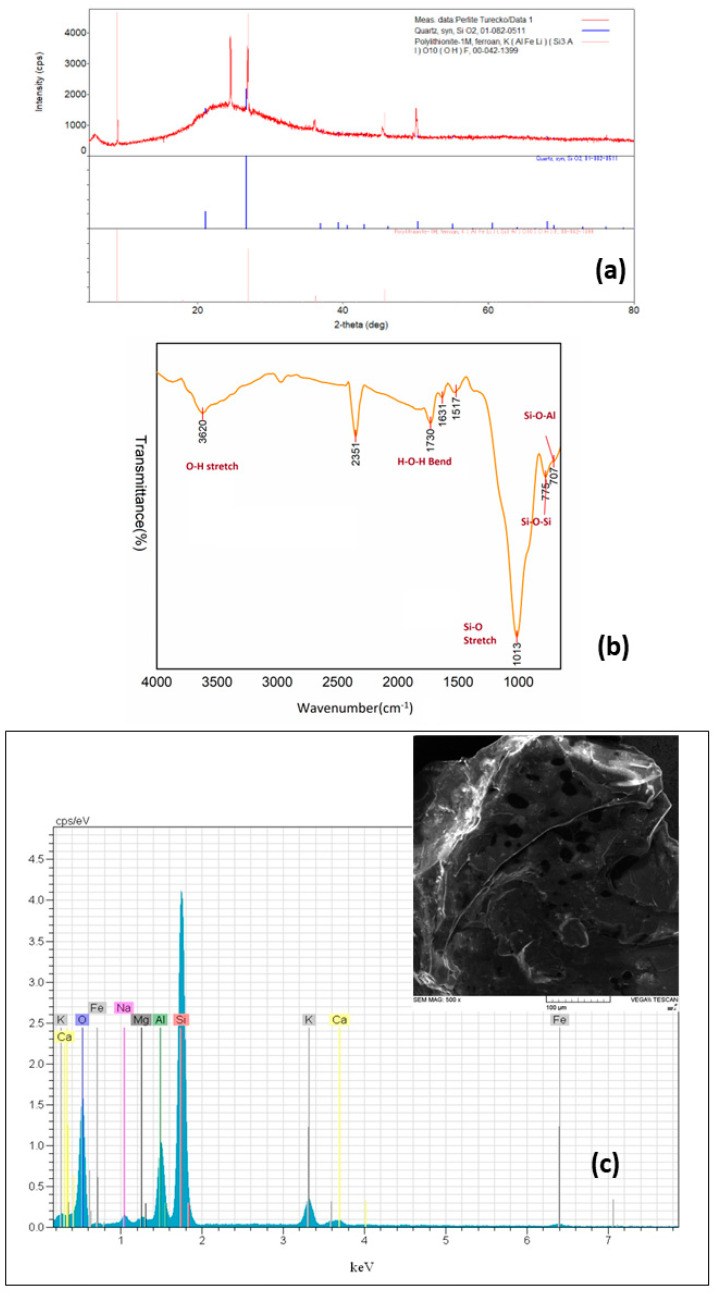
Characterization of perlite: (**a**) XRD pattern; (**b**) FTIR spectrum; (**c**) SEM–EDX analysis.

**Figure 2 materials-15-07816-f002:**
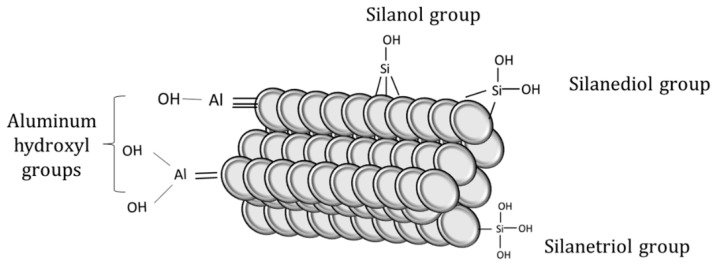
Functional groups on perlite surface.

**Figure 3 materials-15-07816-f003:**
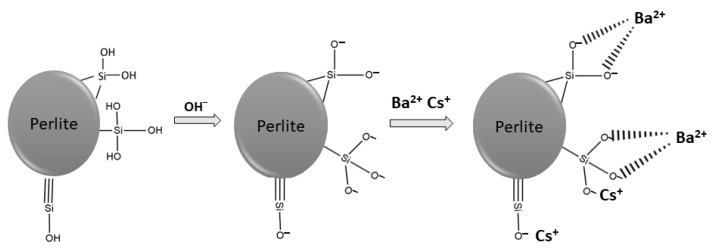
Possible surface reactions and metal bonding mechanism.

**Figure 4 materials-15-07816-f004:**
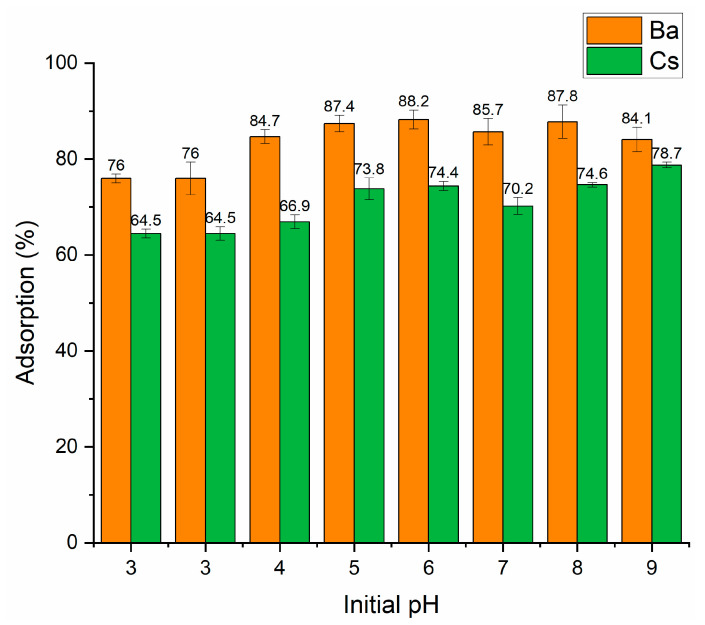
Influence of initial pH on the removal of Ba^2+^ and Cs^+^ using perlite (contact time: 60 min; metal ion concentration: 1 × 10^−^^5^ mol/L; dosage: 10 g/L; temperature: ambient conditions).

**Figure 5 materials-15-07816-f005:**
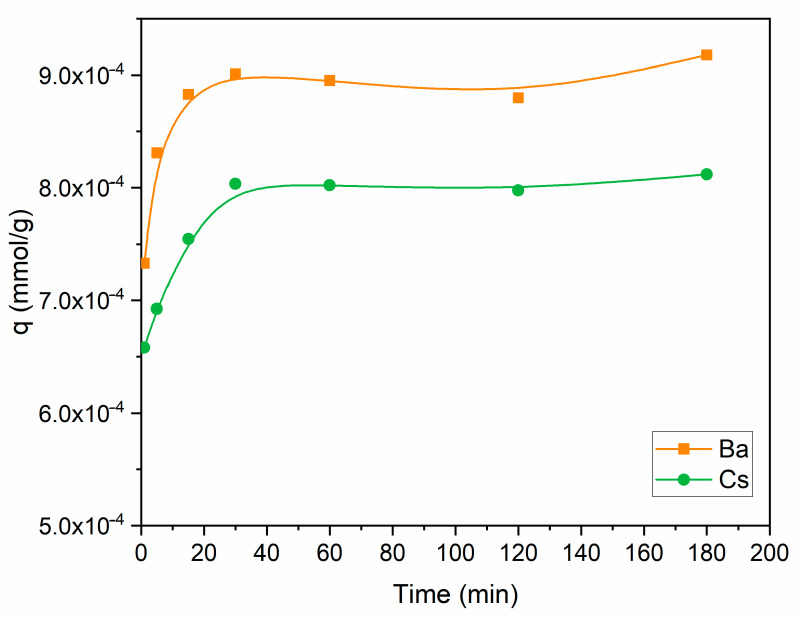
Influence of contact time on the removal of Ba^2+^ and Cs^+^ using perlite (initial pH 6 for Ba^2+^; initial pH 9 for Cs^+^; metal ion concentration: 1 × 10^−^^5^ mol/L; dosage: 10 g/L; temperature: ambient conditions).

**Figure 6 materials-15-07816-f006:**
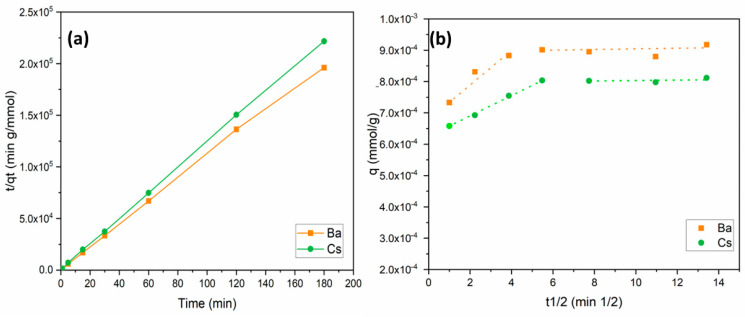
Kinetic models for Ba^2+^ and Cs^+^ removal: (**a**) pseudo-second-order model; (**b**) intraparticle diffusion model.

**Figure 7 materials-15-07816-f007:**
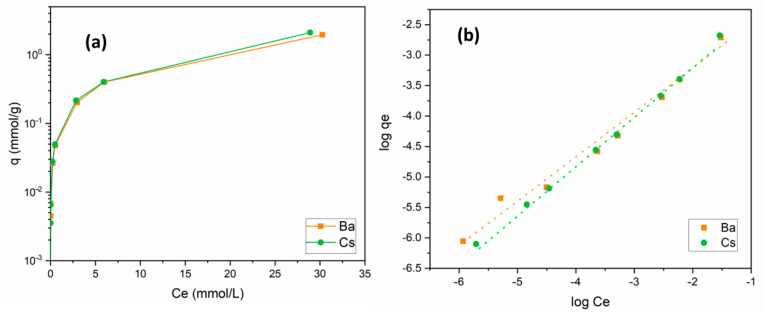
(**a**) Influence of metal ion concentration on the removal of Ba^2+^ and Cs^+^ using perlite; (**b**) Freundlich isotherm (initial pH 6 (Ba^2+^); pH 9 (Cs^+^); contact time: 60 min; dosage: 10 g/L; temperature: ambient conditions).

**Figure 8 materials-15-07816-f008:**
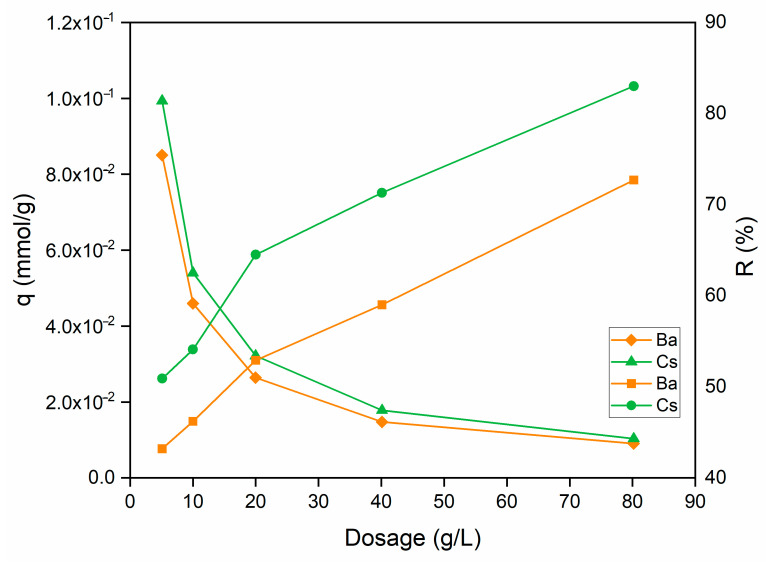
Influence of sorbent dosage on the removal of Ba^2+^ and Cs^+^ using perlite (initial pH 6 (Ba^2+^); pH 9 (Cs^+^); metal ion concentration: 1 × 10^−^^3^ mol/L; contact time: 60 min; temperature: ambient conditions).

**Figure 9 materials-15-07816-f009:**
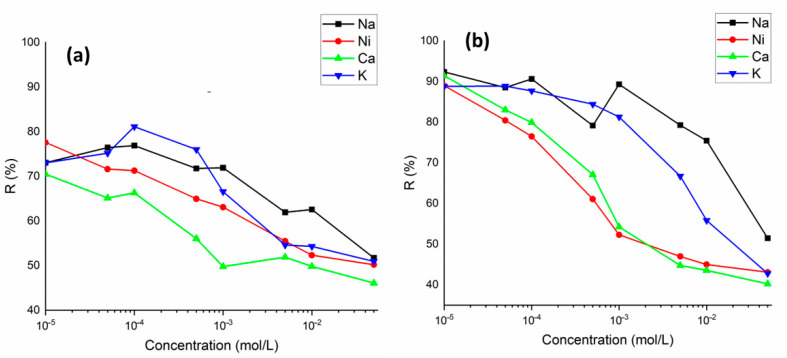
The variation in sorption percentage of Ba^2+^ (**a**) and Cs^+^ (**b**) ions on perlite in the presence of competing ions (Na^+^, K^+^, Ni^2+^, and Ca^2+^) (concentration of Ba^2+^ and Cs^+^: 1 × 10^−5^ mol/L; concentration of competing ions: 1 × 10^−5^ to 5 × 10^−2^ mol/L).

**Figure 10 materials-15-07816-f010:**
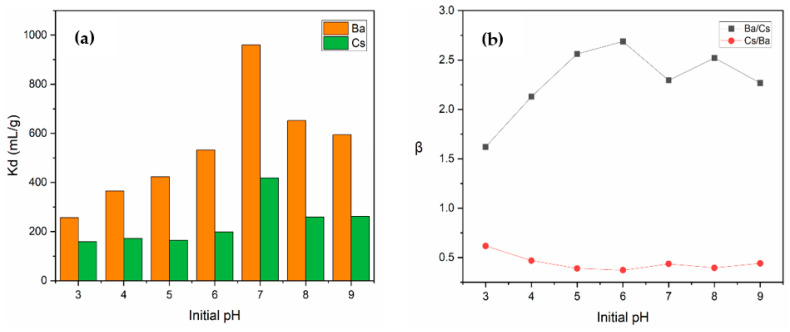
The variation in (**a**) *K_d_* values and (**b**) selectivity coefficients as a function of initial pH (concentration of Ba^2+^ and Cs^+^: 1 × 10^−^^5^ mol/L; contact time: 60 min; temperature: ambient conditions).

**Table 1 materials-15-07816-t001:** Concentration limits of C137s.

Unit (Bq/kg)	EU [4]	USA [5]	Japan [6]
Drinking water	1000	1200	10
Milk	1000	1200	50
General foods	1250	1200	100

**Table 2 materials-15-07816-t002:** Pseudo-first- and second-order model parameters for Ba^2+^ and Cs^+^ sorption onto perlite.

	Pseudo-First-Order			Pseudo-Second-Order		
	*k*_1_ (1/min)	*q_e_* (mmol/g)	*R* ^2^	*k*_2_ (g/mmol min)	*q_e_* (mmol/g)	*R* ^2^
^133^Ba	8.7 × 10^−3^	6.2 × 10^−5^	0.14	1524.11	9.1 × 10^−4^	0.999
^137^Cs	1.9 × 10^−2^	6.9 × 10^−5^	0.45	1655.62	8.1 × 10^−4^	0.999

**Table 3 materials-15-07816-t003:** Intraparticle diffusion model parameters for Ba^2+^ and Cs^+^ sorption onto perlite.

	First Section			Second Section		
	*K_id_*_1_ (mmol/g min^0.5^)	*C*	*R* ^2^	*K_id_*_2_ (mmol/g min^0.5^)	*C*	*R* ^2^
^133^Ba	5 × 10^−5^	7 × 10^−4^	0.936	1 × 10^−6^	9 × 10^−4^	0.069
^137^Cs	3 × 10^−5^	6 × 10^−4^	0.999	2 × 10^−6^	8 × 10^−4^	0.363

**Table 4 materials-15-07816-t004:** Langmuir and Freundlich isotherm model parameters.

Model	Parameters	Ba^2+^	Cs^+^
Langmuir	*q_m_* (mol/g) *b* (L/g) *R*^2^	0.031 44.62 0.322	0.038 34.81 0.317
Freundlich	1/*n* *K_f_* *R*^2^	0.71 0.015 0.979	0.81 0.027 0.996

**Table 5 materials-15-07816-t005:** Comparison of sorption capacity of other natural inorganic sorbents for Ba^2+^ and Cs^+^.

Sorbent	Metal Ion	Metal Concentration (mol/L)	Initial pH	q (mmol/g)	Reference
Iranian expanded perlite	Ba^2+^	3.6 × 10^−5^–3.6 × 10^−4^	6	0.018	[35]
German beidellite	Ba^2+^	3.6 × 10^−5^–1.46 × 10^−3^	6	0.326	[53]
Turkish perlite	Ba^2+^	1 × 10^−5^–5 × 10^−2^	6	1.96	Present study
Argentinian perlite	Cs^+^	3.8 × 10^−5^–3.8 × 10^−3^	5	0.022	[36]
Serbian clinoptilolite	Cs^+^	3.8 × 10^−5^–7.5 × 10^−3^	5	0.369	[54]
Zeolite	Cs^+^	-	7.5	1.48	[55]
Slovak bentonite	Cs^+^	1 × 10^−5^–5 × 10^−2^	7	0.88	[56]
Indian bentonite	Cs^+^	3.8 × 10^−4^–7.5 × 10^−3^	6	1.46	[57]
Turkish bentonite	Cs^+^	0–4 × 10^−2^	8	2.26	[58]
Chabazite	Cs^+^	0–1 × 10^−1^	5	2.07	[59]
Turkish perlite	Cs^+^	1 × 10^−5^–5 × 10^−2^	9	2.11	Present study

## Data Availability

Not applicable.

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
