# Peer review of "Isotherm, Kinetic, and Selectivity Studies for the Removal of 133Ba and 137Cs from Aqueous Solution Using Turkish Perlite"

_materials, 2022, doi:10.3390/ma15217816_

Round 1

Reviewer 1 Report

1.     The main lack of the submitted manuscript is the characterization of perlite. The quantitative elemental composition of Turkish perlite is necessary. How do the authors explain the differences in the elemental analysis obtained with the XRF and XRD analysis? (The elemental composition (identification only) of Turkish perlite was examined by XRF analysis. The most occurred elements were Si, Al, K, Fe, Ni, As, Zn, Rb, Mn, or Zr… The SEM-EDX analysis of perlite is presented in Fig. 1(c). SEM image illustrates the porous structure of the perlite surface. As expected in perlite, Si, Al, O, K, Mg, Na, Ca and Fe elements were determined and data corresponds with the data  provided by XRF analysis.)  How can one explain the presence of the polylithionite crystalline phase, which is characterized by the presence of Li in a larger amount, while Li was not detected in the examined perlite. (The closest possible structures that could belong to a given peak are: quartz and polylithionite.) What does the sentence “The peak around 1652 cm-1 is an indicator of O-H bending stress” mean? Additionally, this band is incorrectly assigned in the Fig. 1b. If this is the band of the bending vibration of water, which it certainly corresponds to in position, where is the band of the stretching OH vibration? Is it in the spectrum of perlite removal of the spectral contributions due to atmospheric water and CO2?  It would be good to specify water content of perlite as hydrated volcanic glass.

2.     It is difficult to read the percentages from Figure 4. 3D columns are beautiful but not practical. To me, from the Figure 4, it is not clear that the maximum sorption percentage is 88.2% for Ba²⁺ at pH. In each case, the values of maximum sorption percentage for Ba²⁺ vary only about 10 % depending on initial pH, in the region 5-8 is actually minimal, which would be convenient for applications in liquid waste streams (in a wider range of pH without adjusting). What are the measurement errors, they need to be added to the Figure?

Author Response

Answers to Reviewer 1

 Thank you very much for your comments, please below you can find the answer onto each comment separately. The manuscript was improved, and the changes were colored by yellow.  The English was checked by native speaker and some proves were provided.

 Q1: The main lack of the submitted manuscript is the characterization of perlite. The quantitative elemental composition of Turkish perlite is necessary. How do the authors explain the differences in the elemental analysis obtained with the XRF and XRD analysis? (The elemental composition (identification only) of Turkish perlite was examined by XRF analysis. The most occurred elements were Si, Al, K, Fe, Ni, As, Zn, Rb, Mn, or Zr… The SEM-EDX analysis of perlite is presented in Fig. 1(c). SEM image illustrates the porous structure of the perlite surface. As expected in perlite, Si, Al, O, K, Mg, Na, Ca and Fe elements were determined and data corresponds with the data  provided by XRF analysis.)  How can one explain the presence of the polylithionite crystalline phase, which is characterized by the presence of Li in a larger amount, while Li was not detected in the examined perlite. (The closest possible structures that could belong to a given peak are: quartz and polylithionite.)

A1: Thank you very much for your comment. Unfortunately, the XRF device used is not able to identify elements with a proton number less than approx. 10. Therefore, Li can never be identified by our device. The XRD spectrum indicates that the given perlite has a structure similar to polylithionite - but this does not mean that polylithionite is actually found in perlite.

Q2: What does the sentence “The peak around 1652 cm-1 is an indicator of O-H bending stress” mean? Additionally, this band is incorrectly assigned in the Fig. 1b. If this is the band of the bending vibration of water, which it certainly corresponds to in position, where is the band of the stretching OH vibration? Is it in the spectrum of perlite removal of the spectral contributions due to atmospheric water and CO2?  It would be good to specify water content of perlite as hydrated volcanic glass.

A2: Thank you very much for comment. The FTIR spectrum and the explanation of bonding-sides were revised in manuscript (L 204)

Q3: It is difficult to read the percentages from Figure 4. 3D columns are beautiful but not practical. To me, from the Figure 4, it is not clear that the maximum sorption percentage is 88.2% for Ba²⁺ at pH. In each case, the values of maximum sorption percentage for Ba²⁺ vary only about 10 % depending on initial pH, in the region 5-8 is minimal, which would be convenient for applications in liquid waste streams (in a wider range of pH without adjusting). What are the measurement errors, they need to be added to the Figure?

A3: Thank you very much for comments, as you suggested, the error bars were inserted and the Fig. 4, was revised (L 231).

Reviewer 2 Report

1-    Regarding the radiation caused by Ba and Cs, is it safe to work with them in the laboratory? If it is not completely safe, how it was managed?

2-    Error bars are needed for fig. 4 to compare the effect of different pH values exactly.

Author Response

Answers to Reviewer 2

Thank you very much for your comments, please below you can find the answer onto each comment separately. The manuscript was improved, and the changes were colored by yellow.  The English was checked by native speaker, and some proves were provided.

Q1:  Regarding the radiation caused by Ba and Cs, is it safe to work with them in the laboratory? If it is not completely safe, how it was managed?

A1: Thank you very much for your comment, the experimental date was provided at Department of Nuclear Chemistry, Faculty of Natural Sciences, Comenius university in Bratislava. The department operates several laboratories, where is possible to work with radiation. Laboratories are in special category divided and categorized by the Slovak law. To work in this laboratory is possible under the special regulations, about high radiation hygiene, safety, and health protection during the experimental work.

Q2:  Error bars are needed for fig. 4 to compare the effect of different pH values exactly.

A2: Thank you very much for comment, as you suggested, we inserted error bars, and the Fig. 4 was revised (L 231).

Reviewer 3 Report

This article reports on the adsorption and removal of radioactive isotopes 133 Ba and 137 Cs by using natural perlite as adsorbent. The study is interesting both from environmental point and from methodological, given the rarity of use of radioanalytical methods in determining sorption characteristics. The manuscript is well written and is publishable in Materials after incorporating the following changes/ modifications.

Major corrections.

1. The sorption capacities of similar and related materials in other studies, as shown in Table 5, seem to be 1-2 orders of magnitude lower, than the results of the present study.

It seems unlikely that this particular material had such specific characteristics, so the differences can be searched in the different conditions used in those studies, or, eventually, some artefacts. Authors should provide a full discussion on this point, and extend the Table with some other materials, as well as include the experimental conditions of sorption capacity determination.

2. There is no any discussion about the mechanism of the sorption, which surface functional groups can be involved in it.

3. There are no recovery experiments presented.

Minor corrections.

Some figures need to be replaced with higher quality/resolution versions.

Author Response

Answers to Reviewer 3

Thank you very much for your comments, please below you can find the answer onto each comment separately. The manuscript was improved, and the changes were colored by yellow.  The English was checked by native speaker, and some proves were provided.

Q1: The sorption capacities of similar and related materials in other studies, as shown in Table 5, seem to be 1-2 orders of magnitude lower, than the results of the present study. It seems unlikely that this particular material had such specific characteristics, so the differences can be searched in the different conditions used in those studies, or, eventually, some artefacts. Authors should provide a full discussion on this point, and extend the Table with some other materials, as well as include the experimental conditions of sorption capacity determination.

A1: Thank you for your comment. Table 5 was improved by adding new materials and providing experimental conditions. We obtained relatively higher uptake capacities due to the higher initial metal concentrations. Similar uptake values were also found with zeolite, Indian bentonite, Turkish bentonite and chabazite (L 346)

Q2: There is not any discussion about the mechanism of the sorption, which surface functional groups can be involved in it.

A2: Thank you very much for your comment. The section 3.2.1 and Fig.3 provide data on the possible sorption mechanism. Surface groups that may be responsible for sorption were mentioned (Line 215).

Q3: There are no recovery experiments presented.

A3: Thank you very much for your suggestion. The recovery experiments were not our main target within the context of the present work. However, further research can include desorption studies.

Q4:  Some figures need to be replaced with higher quality/resolution versions.

A4: Thank you very much for your suggestion, the Fig. 1, Fig. 4 and Fig. 10 were proved.

Round 2

Reviewer 1 Report

The revised version is a little bit improved, but still possesses several lacks.

It's a pity the authors can't do a better characterization of the material. This remains the manuscript's greatest shortcoming.

Figure 3: Ba+2 must be change into Ba2+

Table 5 is a new one, and needs to be rearranged. Recalculate the cited results (all into mol/L or mg/L, to be comparable) and group them according to ions, to be easier to follow.

Author Response

Answers to Reviewer 1

Thank you very much for your time and constructive comments, please below you can find the answer onto each comment separately. Unfortunately, we only have access to characterization mentioned in manuscript XRD, XRF, SEM, EDX, FTIR and BET.

Q1: Figure 3: Ba+2 must be change into Ba2+

A1: Thank you very much for your comment. The required revision in Figure 3 was made (Line 230).

Q2: Table 5 is a new one and needs to be rearranged. Recalculate the cited results (all into mol/L or mg/L, to be comparable) and group them according to ions, to be easier to follow.

A2: Thank you very much for comment. Metal concentration values were converted into mol/L and data were grouped according to ions (Line 346).

 Authors

Reviewer 3 Report

The comments have been properly addressed.

Author Response

Answers to Reviewer 3

Thank you very much for your time and constructive comments.

Authors
